# Compound Effects of Sodium Chloride and Gypsum on the Compressive Strength and Sulfate Resistance of Slag-Based Geopolymer Concrete

**Wei He [1], Benxiao Li [1], Xia Meng [2],\* and Quan Shen [3]**

[1]  School of Civil and Transportation Engineering, Beijing University of Civil Engineering and Architecture, Beijing 100044, China

[2]  Architectural Design and Research Institute of Tsinghua University Co., Ltd., Beijing 100084, China

[3]  Shanghai Interlink Road & Bridge Engineering Co., Ltd., Shanghai 201213, China

\*   Correspondence: mengxia@thad.com.cn

**Abstract:** Based on compressive strength, sulfate resistance, mass change, and relative dynamic elastic modulus tests, and XRD and SEM analysis, the effects of sodium chloride (NaCl) and gypsum on the mechanical properties and resistance to sulfate attack of slag-based geopolymer concrete activated by quicklime as well as the mechanism of action were studied. The results indicate that: (1) with appropriate dosages of NaCl or gypsum, the compressive strength of geopolymer concrete can be increased by 55.8% or 245.3% at 3 days and 23.9% or 82.3% at 28 days, respectively. When NaCl and gypsum are combined, Friedel's salt, Kuzel's salt, and NaOH are generated, and the strength is increased by 90.8% at 3 days, and 180.3% at 28 days. (2) With 2% NaCl alone, the mass loss is reduced from 5.29% to 2.44%, and the relative dynamic elastic modulus is increased from 0.37 to 0.41. When compounded with 7.5% gypsum, the mass is increased by 0.26%, and the relative dynamic elastic modulus is increased to 1.04. With a further increase of NaCl to 4%, the mass is increased by 0.27%, and the relative dynamic elastic modulus is increased to 1.09. The sulfate corrosion resistance coefficient of geopolymer concrete is increased from 0.64 to 1.02 when it is immersed with 7.5% gypsum alone for 90 days, and it can be further increased to 1.11 when compounded with 4% NaCl. (3) The geopolymer prepared with sodium chloride: gypsum: quicklime: slag = 4:7.5:13.5:75 can be used to replace 32.5 slag Portland cement in plain concrete. The cost and carbon emissions are reduced by 25% and 48%, respectively, and the sulfate corrosion resistance coefficient is higher by 38.8% than with slag Portland cement.

**Keywords:** NaCl; gypsum; slag-based geopolymer; compressive strength; sulphate attack

## 1. Introduction

Cement is one of the main raw materials for the building industry. The production of Portland cement consumes a large amount of natural resources, has very high energy consumption, and causes significant environmental pollution, including high $CO_2$ emissions resulting from the decarbonization of raw material. The comprehensive energy consumption of cement clinker production and cement grinding is up to 120 kg standard coal/t and 30 kWh/t [1,2], respectively; the carbon dioxide emission is up to 830 kg/t [3]. Consequently, carbon emissions from the cement industry account for about 8% of global anthropogenic carbon dioxide emissions [4,5]. In addition, $NO_x$, $SO_2$, and other harmful gases are also emitted during the cement production process [6,7]. In 2021, worldwide cement output was 4.31 billion tons, with China contributing about 55% of the total (2.38 billion tons) [8,9]. As of 2020, among the top 15 carbon-emitting countries, 10 countries have achieved their carbon emission peak, including the United States, Japan, Germany, and France. In 2020, the Chinese government pledged to peak carbon emissions by 2030 and achieve carbon neutrality by 2060 [10]. However, China's cement industry

carbon emissions in 2020 were as high as 1.466 billion tons [11], making them the second largest source of carbon emissions from manufacturing [12]. The reduction of carbon emissions in the cement industry is an important factor to allow China to achieve the 'double carbon' target. Alkali-activated geopolymers are green new cementitious materials that do not require "two grinding and one burning" and can utilize a large number of industrial byproducts, such as slag, fly ash, and electric furnace phosphorus slag [13,14], and these geopolymers have properties that are comparable to cement. Silica fume can also be used to improve the properties of geopolymer concrete [15]. However, it is usually necessary to use strong alkaline activators, such as caustic soda or caustic potash [16,17], water glass [18,19], etc., to create conditions for the dispersion and dissolution of slag vitreous and promote the continuous hydration reaction of $SiO_2$ and $Al_2O_3$ in the mixture [20]. The comprehensive power consumption of caustic soda and caustic potash production reaches 2427 kWh/t [21] and 2150 kWh/t [22], respectively; in addition, the process of water glass production consumes caustic soda, causing additional electric energy costs. Since China's electricity mainly comes from thermal power generation, the carbon reduction ability of geopolymers activated by strong alkali is restricted.

Quicklime can react with water and produce $Ca(OH)_2$. The activation process when using quicklime to activate slag is too long to obtain ideal effects, however, making it difficult to use in engineering. Therefore, we attempted using building gypsum and sodium chloride to assist the activation to improve the activation efficiency of slag and make it applicable to engineering. Studies [23,24] have shown that slag-based geopolymer activated by NaCl, building gypsum, and quicklime can meet the strength requirements of 32.5 slag Portland cement under the appropriate dosage. This geopolymer has the advantage of high early strength, and it also avoids high shrinkage [13] because of the hydration's product's high crystal composition. Moreover, it has the advantages of low energy consumption, low carbon emission, and low cost. The producing energy consumption of quicklime, building gypsum, and slag powder is 147 kgce/t [25], 43 kgce/t [26], and 16 kgce/t [27], respectively. Therefore, the energy consumption of the slag-based geopolymer calculated by the dosage is only about 35 kgce/t, while the energy consumption of the slag-based geopolymer activated by strong alkali activators reaches 130 kgce/t. The carbon emission of slag powder, quicklime, and building gypsum is about 60 kg/t [28], 368 kg/t [29], and 200 kg/t [30], respectively, so the carbon emission of composite activated geopolymer is about 110 kg/t, which is about 75% lower than that of 32.5 slag cement. The geopolymer activated by the composite activator consumes less energy and emits less carbon dioxide. Moreover, the price of the strong alkali-activated geopolymer and 32.5 slag Portland cement is about 500 yuan/t and 400 yuan/t, respectively, while the price of the geopolymer activated by composite activators is only 300 yuan/t when calculated according to the market price, which is 40% lower than that of the strong alkali-activated geopolymer and 25% lower than 32.5 slag Portland cement.

In the slag-based geopolymer activated by composite activators consisting of NaCl, building gypsum, and quicklime, quicklime hydrates with water and produces $Ca(OH)_2$, which can form a certain alkaline environment and promotes the hydration of slag [31,32]; however, the slow hydration results in low early strength. The added gypsum can activate the slag activity and generate ettringite, improving early strength [33–35]; however, ettringite may harm to strength [36–39]. NaCl also has the effect of improving strength [40–45]. Chloride ions can also react with AFm and C-A-H to form Friedel's salt [46,47]. There are few studies on the compound effects of sodium chloride and gypsum on slag-based geopolymer resulting from the combination of NaCl and gypsum. At the same time, in the service environment of sulfate, the sulfate erosion induced by AFT may be aggravated for the geopolymer. Hence, this paper studied the compound effects and mechanism of NaCl and gypsum on the compressive strength, the resistance to water, and the resistance to sulfate attack of geopolymers.

## 2. Materials and Methods

### 2.1. Materials

The S95 slag powder utilized in this paper was manufactured by Hebei Jintaicheng Environmental Resources Co, Ltd., with a density of 2.87 g/cm$^3$. Chemical compositions, specific surface area, and particle size distribution are shown in Tables 1 and 2 and Figure 1, respectively. The slag's chemical module K = 2.22; alkalinity modules Mo = 1.10; the observed 28 d activity coefficient was 95%. The chemical composition of the slag is analyzed by XRF, and the particle size distribution is analyzed by Mastersizer 3000.

**Table 1.** Chemical composition of slag (mass %).

| SiO$_2$ | Al$_2$O$_3$ | SO$_3$ | MgO | CaO | Fe$_2$O$_3$ | Na$_2$O | K$_2$O |
|---------|-------------|--------|------|-------|-------------|---------|--------|
| 28.65 | 16.49 | 2.20 | 9.04 | 40.86 | 0.23 | 0.46 | 0.48 |
| **Cl** | **P$_2$O$_5$** | **TiO$_2$** | **MnO** | **SrO$_2$** | **Y$_2$O$_3$** | **ZrO$_2$** | **Others** |
| 0.04 | 0.05 | 0.78 | 0.52 | 0.13 | 0.01 | 0.05 | 0.01 |

**Table 2.** Particle size distribution of slag.

| Specific Surface Area | Dx (10) | Dx (50) | Dx (90) |
|-----------------------|---------|---------|---------|
| 432.9 m$^2$/kg | 2.23 μm | 12.0 μm | 28.3 μm |

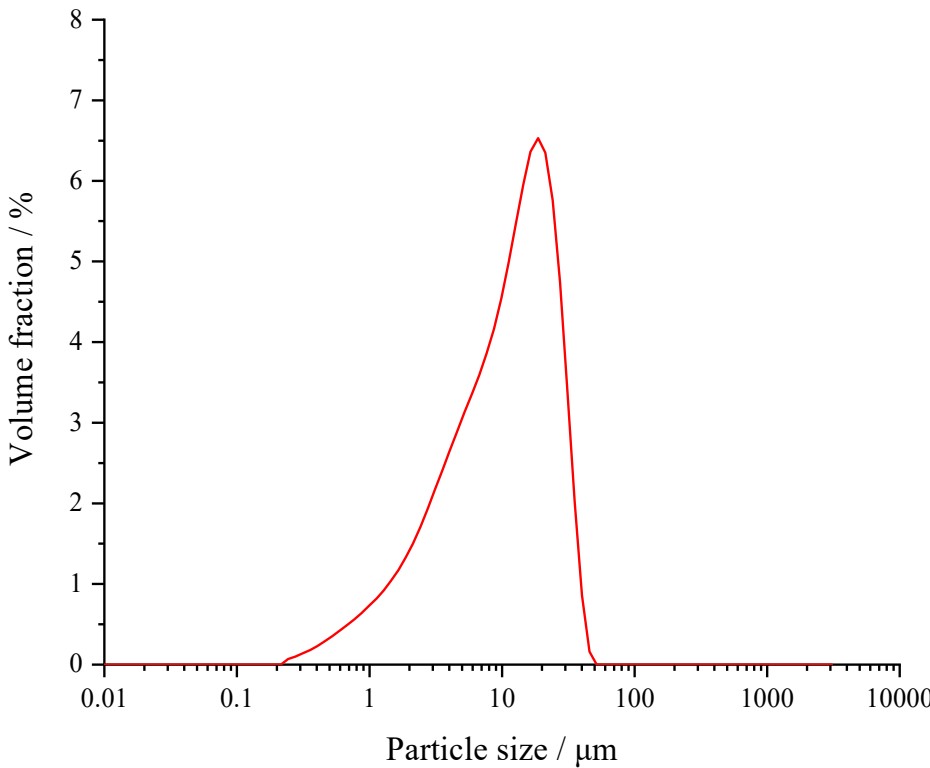

**Figure 1.** Particle size distribution curve.

The quicklime used in the test was produced by Jiangxi Chuangxian Fine Calcium Industry Co., Ltd., with CaO content over 95%. NaCl was produced by Jiangxi Jinghao Salt Chemical Co., Ltd., with NaCl content of 99.1%. Building gypsum ($CaSO_4 \cdot 0.5H_2O$) was produced by Jinan Yuxin Chemical Co., Ltd. The cement used in the reference group was 32.5 slag Portland cement produced by Zhucheng Yangtai Cement Co., Ltd. The materials used in the geopolymer are shown in Figure 2. The fine aggregate was natural

river sand, the fineness modulus was 2.7, the apparent density was 2630 kg/m$^3$, and the coarse aggregate was gravel with a particle size of 5–20 mm, and apparent density of 2780 kg/m$^3$. The absorption of fine aggregate and coarse aggregate was 1.0% and 0.7%, respectively. Mixing water used in this study came from municipal tap water in Beijing, China, which met drinking water quality requirements.

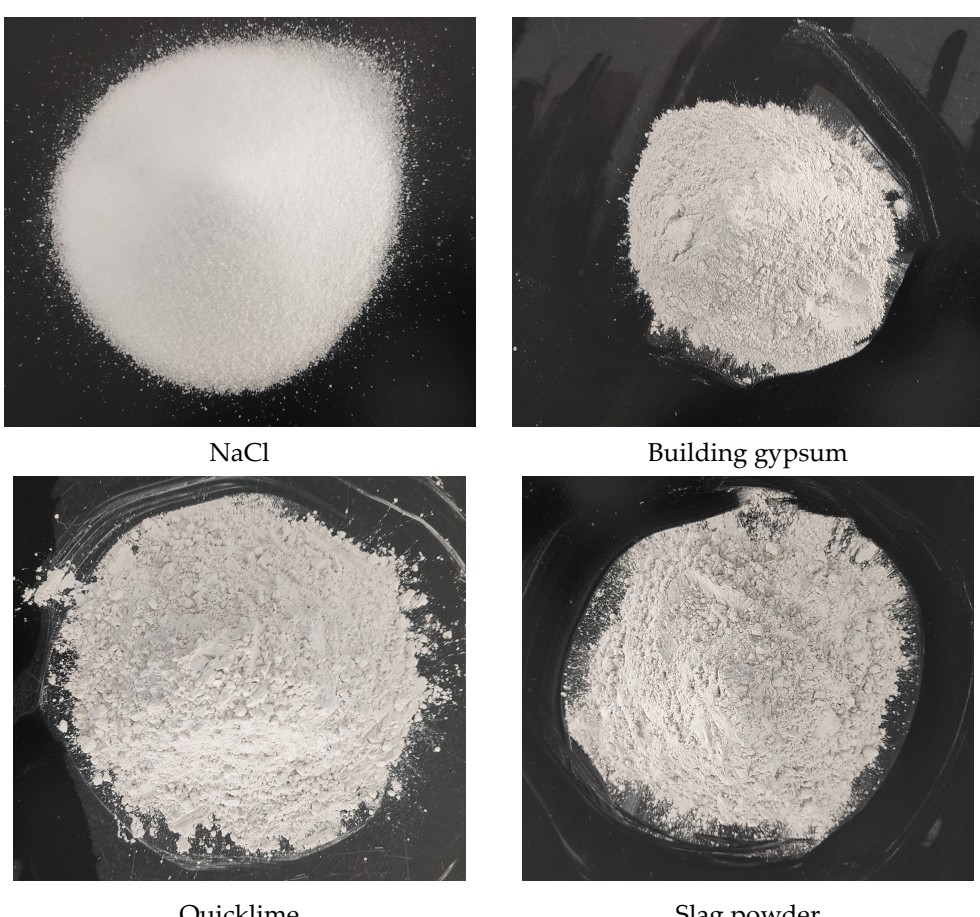

| NaCl | Building gypsum |
| Quicklime | Slag powder |

**Figure 2.** Materials used in geopolymer.

The flowchart of the study is shown in Figure 3.

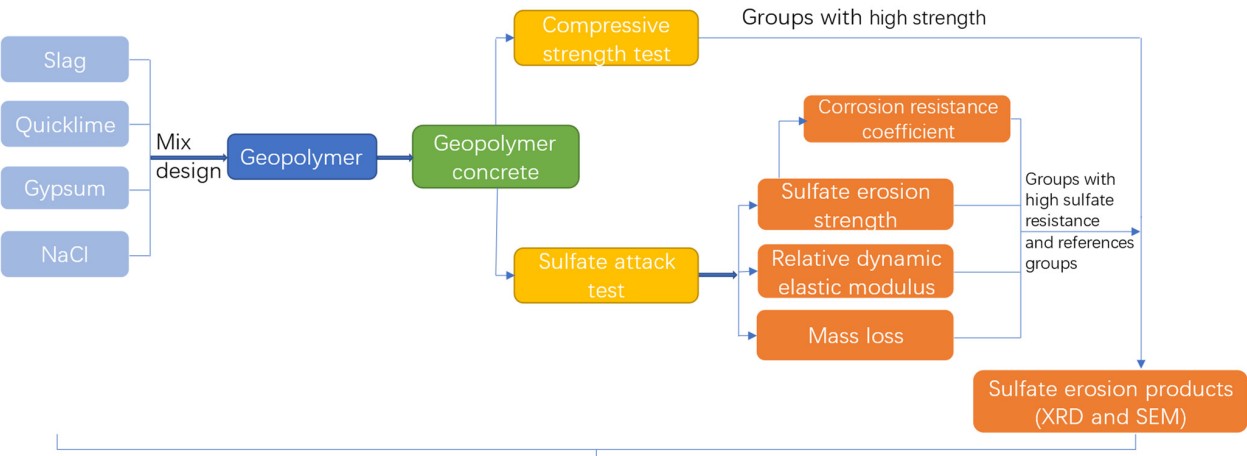

**Figure 3.** Flowchart of the research.

### 2.2. Experiment Methods

(1) Mix design

According to the author's previous research results [23], the 28-day compressive strength of geopolymer mortar is similar to that of 32.5 slag Portland cement mortar when NaCl: magnesite: building gypsum: quicklime: slag = 2%:8%:10%:5%:75%. The main function of magnesite is to reduce the shrinkage of geopolymers [48], while the price of magnesite is relatively high. Consider optimizing the components by removing magnesite and adding more calcium hydroxide to further promote the reaction between gypsum and slag to generate more AFt, thereby compensating for shrinkage. Further mortar tests show that when NaCl: gypsum: quicklime: slag = 2%:7.5%:15.5%:75%, the 3-day and 28-day compressive strength of geopolymer mortar can still reach this level, and the price is even lower. In order to apply these results to plain concrete, the C 30 concrete mix ratio was designed as geopolymer: coarse aggregate: fine aggregate: water = 471:1050:700:179 based on the 28-day compressive strength of slag-based geopolymer mortar measured, with a 0.38 water–binder ratio, according to 'Ordinary Concrete Mix Proportion Design Regulations'(JGJ55-2011).

Durability properties play a key role in controlling the longer service life of concrete [49,50], and sulfate attack on concrete is one of the most complex and harmful types of environmental water erosion, combining physical and chemical effects. The erosion involves the transmission of sulfate ions in concrete, the reaction between sulfate ions and cement hydration products, and the destruction of concrete structure by the generated expansion products [51–53]. In order to avoid stability problems caused by sulfate attack, 'Common Portland cement' (GB 175-2007) stipulates that $SO_3$ in slag Portland cement should not exceed 4% and other cement should not exceed 3.5%, but the amount of $SO_3$ used in the geopolymer concrete far exceeds this limit; the amount of $SO_3$ is actually higher than the content of gypsum in many other geopolymers. Such a large amount of internal gypsum has a high risk of sulfate erosion under the compound action of an external high-sulfate environment. In addition, the calcium-hydroxide activator used can very easily induce a sulfate attack. No cracking problems are caused by swelling in the conventional curing test, but whether its mechanical properties will be seriously affected when there are sulfates in the environment is worthy of further study. When the environment contains $SO_4^{2-}$, the external $SO_4^{2-}$ may invade the interior of the concrete, then undergo a series of chemical reactions with hydration products such as C-A-H, $Ca(OH)_2$, and AFm in the concrete to form gypsum [54], AFt [55], etc. These products can cause concrete damage due to volume expansion as well as decomposing or dissolving the hydration products of cement $Ca(OH)_2$ and C-S-H gel, etc.; either of these activities reduces the strength and bonding properties of concrete.

In order to evaluate the influence of NaCl and gypsum components in the slag-based geopolymer on its compressive strength and sulfate corrosion resistance under the condition that the composite activator component remains unchanged at 25%, the Friedel's salt generated by the participation of NaCl in the reaction is considered. Friedel's salt can react with $SO_4^{2-}$ to form Kuzel's salt [56–58], meaning that the hardened paste has a better ability to accommodate sulfate. The NaCl content is there increased by 4% and 6% on the basis of 2%. In addition, when there is $SO_4^{2-}$ in the environmental medium, the products involved in later hydration have a certain blocking and filling effect on the pores [59,60], thus hindering the further entry of $SO_4^{2-}$ in the environment, and the optimal amount of gypsum may change. Therefore, based on fluctuations in gypsum content from 7.5% to 10% and from 5% to 2.5% was also set, and experimental groups 1–12 were set accordingly. In order to characterize the influence degree of NaCl and gypsum under a condition where the amount of slag and the ration of the activator are unchanged, control groups 13 to 15 were created separately; in these groups, NaCl was 0, the gypsum was 0, and the NaCl and gypsum were both 0, respectively. To compare with slag cement, experimental group 16 was set up. These groups are shown in Table 3.

**Table 3.** Mixture proportions of binders /%.

| No. | Sample | NaCl | Building Gypsum | Quicklime | Slag |
|---|---|---|---|---|---|
| 1 | LA | | 2.5 | 20.5 | 75 |
| 2 | LB | 2 | 5 | 18 | 75 |
| 3 | LC | | 7.5 | 15.5 | 75 |
| 4 | LD | | 10 | 13 | 75 |
| 5 | MA | | 2.5 | 18.5 | 75 |
| 6 | MB | 4 | 5 | 16 | 75 |
| 7 | MC | | 7.5 | 13.5 | 75 |
| 8 | MD | | 10 | 11 | 75 |
| 9 | HA | | 2.5 | 16.5 | 75 |
| 10 | HB | 6 | 5 | 14 | 75 |
| 11 | HC | | 7.5 | 11.5 | 75 |
| 12 | HD | | 10 | 9 | 75 |
| 13 | 0C | 0 | 8 | 17 | 75 |
| 14 | L0 | 3 | 0 | 22 | 75 |
| 15 | OQ | 0 | 0 | 25 | 75 |
| 16 | R | | 32.5 slag Portland cement | | |

According to the mixing ratio in Table 3, 27 geopolymer concrete blocks of 100 mm × 100 mm × 100 mm were prepared for each group, including 6 concrete blocks for 3-day and 28-day compressive strength, and 18 concrete blocks for water immersion tests and sulfate immersion tests. After standard curing for 28 days, half of the 18 concrete blocks were immersed in water and the other half in sulfate solution for sulfate attack testing. The erosion strength of immersion was tested for 28 d, 56 d, and 90 d. In addition, the mass and relative dynamic elastic modulus of the three concrete blocks used in the 90-day immersion compressive strength condition were also tested. The mass concentration of $SO_4^{2-}$ in common groundwater in China can be as high as 5 g/L [61], and the mass concentration of $SO_4^{2-}$ in groundwater or saline soil in coastal and individual inland areas can be as high as 50 g/L [62]. To enhance the erosion efficiency, 100 g/L sodium sulfate solution was used as the corrosion solution. The test items were shown in Table 4. Because the immersion test uses a new count of immersion days, the '0 d' of immersion age is set the same as standard curing for 28 d. The compressive strength test is shown in Figure 4.

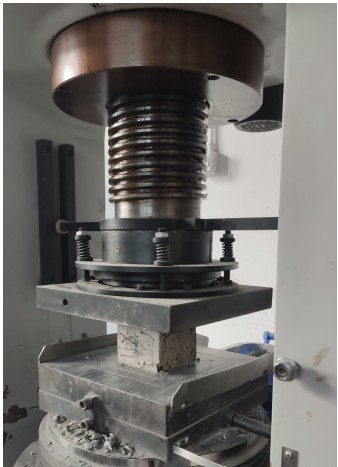

**Figure 4.** Compressive strength test.

**Table 4.** Experimental items.

| No. | Sample | Standard Curing | | Water/Sulfate Attack Test | | | | Erosion Products Analysis of 90 d |
|---|---|---|---|---|---|---|---|---|
| | | Compressive Strength | | Compressive Strength after Immersion | Mass Change | | Relative Dynamic Elastic Modulus | |
| | | | | | Immersion Age | | | |
| | | 3 d | 28 d | 0 d | 28 d | 56 d | 90 d | |
| 1 | LA | | | | | | | / |
| 2 | LB | | | | | | | / |
| 3 | LC | | | | | | | √ |
| 4 | LD | | | | | | | / |
| 5 | MA | | | | | | | / |
| 6 | MB | | | | | | | / |
| 7 | MC | | | | | | | √ |
| 8 | MD | √ | √ | √ | √ | √ | √ | √ |
| 9 | HA | | | | | | | / |
| 10 | HB | | | | | | | / |
| 11 | HC | | | | | | | / |
| 12 | HD | | | | | | | / |
| 13 | 0C | | | | | | | √ |
| 14 | L0 | | | | | | | √ |
| 15 | OQ | | | | | | | √ |
| 16 | Reference | | | | | | | / |

(2) Sulphate attack

The sulfate corrosion resistance of geopolymer concrete was tested through the total immersion experiments. In order to evaluate the damage degree of geopolymer materials under sulfate attack, use the non-metallic ultrasonic testing analyzer to measure the ultrasonic propagation velocity [63] and convert the propagation sound velocity into relative dynamic elastic modulus to non-destructively determine the internal damage of materials [64–67]. During the ultrasonic measurement, the counter-measurement method was used to measure from the center point on both sides of the test block. The selected non-metallic ultrasonic analyzer has a measuring sensor frequency of 5 kHz and a measuring length of 100 mm. During the surface test, a small amount of petroleum jelly can be applied to the probe to ensure that it is tightly connected to the surface of the test piece. The instrument used is shown in Figure 5.

The relative dynamic elastic modulus can be calculated according to the following formula:

$$E_d = \frac{(1+v)(1-2v)\rho v^2}{1-v} = \frac{(1+v)(1-2v)\rho L^2}{1-v} \tag{1}$$

$$E_{rd} = \frac{E_{dn}}{E_{d0}} = \frac{V_n^2}{V_0^2} = \frac{t_0^2}{t_n^2} \tag{2}$$

$E_d$—dynamic elastic modulus $(GPa)$
$E_{rd}$—relative dynamic elastic modulus
V—ultrasonic velocity $(m/s)$
$\rho$—density of concrete $(kg/m^3)$
$v$—Poisson ratio
$V_0$—ultrasonic velocity of specimen before erosion $(m/s)$

$V_n$—ultrasonic velocity of specimen after n days immersion ($m/s$)
$t_0$—sound time before erosion ($\mu s$)
$t_n$—sound time after n days of erosion ($\mu s$)

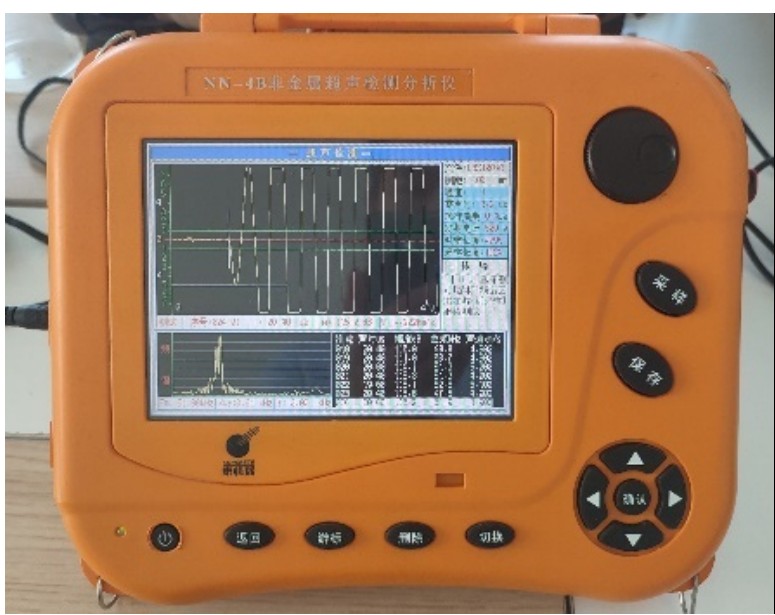

**Figure 5.** NN-4 B non-metal ultrasonic tester.

The test of mass loss after sulfate attack is shown in Figure 6.

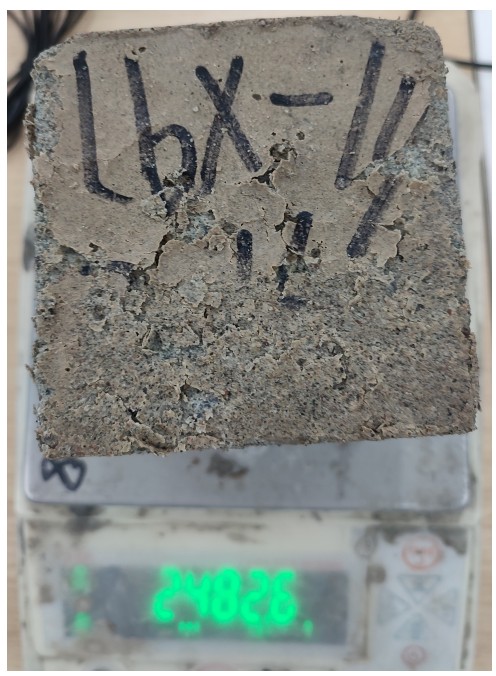

**Figure 6.** Mass loss test after sulfate attack.

The calculation method of anti-corrosion coefficient *K* (>0.8 is qualified) and mass loss is as follows.

$$\text{Mass loss}: \ W_s = \frac{W_b - W_h}{W_b} \tag{3}$$

$$\text{Corrosion resistance coefficient}: \ K = \frac{R_2}{R_1} \tag{4}$$

$W_b$—mass before immersion
$W_h$—mass after immersion
$R_2$—compressive strength in sodium sulfate solution
$R_1$—compressive strength in water

(3) XRD and SEM analysis

The concrete surface samples immersed in sulfate solution for 90 days were taken as reference groups RQ, L0, 0C, and LC, MC, and MD with higher strength. Removing coarse aggregate, these samples were dried at 60 degrees Celsius for 24 h, and then ground for XRD phase analysis to determine the hydration products. The instrument used was Ultima IV, Japan, 3 kW power, a scanning range of 5 °−60 °, scanning speed of 10 °/min. To verify the erosion products, the dried small block samples taken from the concrete surface were analyzed by SEM to observe the morphology of the products. The instruments used were the Hitachi S-3400 scanning electron microscope.

## 3. Results and Analysis

*3.1. Compound Effect of NaCl and Gypsum on the Compressive Strength of Geopolymer Concrete*

The compressive strengths of geopolymer concrete after standard curing for 3 days and 28 days are shown in Figures 7 and 8.

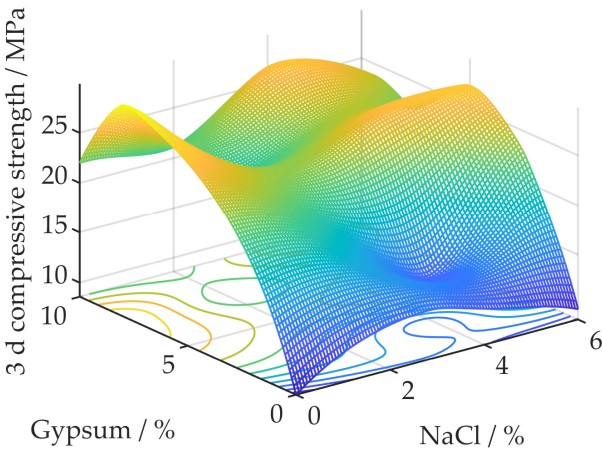

**Figure 7.** 3-day compressive strength of concrete.

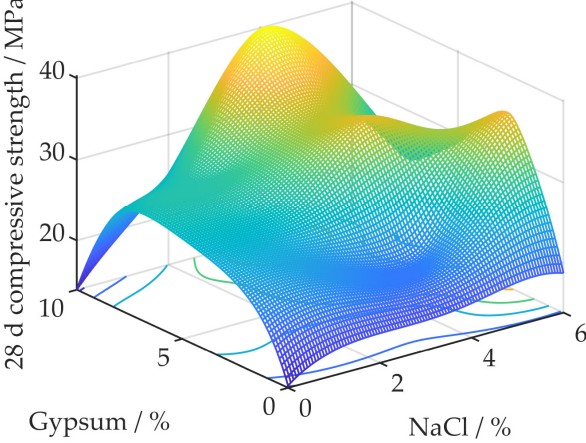

**Figure 8.** 28-day compressive strength of concrete.

As shown in Figure 7, when NaCl or gypsum is used alone, the 3-day compressive strengths both show a single-peak curve. NaCl can improve the 3-day compressive strength at low dosages. The 3-day compressive strength of L0 reaches 13.4 MPa, increased by 55.8% compared to the OQ group. The reason for this phenomenon is that the ionic radius of Na$^+$

is small, making it easy to penetrate into the solid–liquid mixture paste and balance the negative charge generated by the aluminum–oxygen tetrahedron. The Zeta potential in the solution can be reduced by $Cl^-$ [68], which is beneficial to the polycondensation of silicon–aluminum–oxygen tetrahedral units. In addition, sodalite ($Na_4Al_3Si_3O_{12}Cl$) with the same β-characteristic cage as zeolite is produced due to the addition of NaCl. The coplanar connection property makes sodalite a stable structure [69,70]. However, a further increase of NaCl results in a slight strength decrease. When the $Na^+$ concentration is too high, a 'passivation effect' [71] occurs that is caused by the combination of $Na^+$ in the solution with oxygen atoms on the aluminosilicate solid surface. This effect can hinder the erosion and dissolution of the solid raw material by the alkali and inhibit the hydration of the slag [72]. When gypsum is used alone, it has an obvious promoting effect on the 3-day compressive strength of geopolymer concrete; with the increase of gypsum content, this promoting effect tends to increase first and then weaken. The 3-day compressive strength of the 0C group reaches 29.7 MPa, which is 245.3% higher than that of the 00 group. This is mainly because gypsum reacts with C-A-H and generates a large amount of AFt [73] in the early stage of hydration. In addition, the consumption of active $Al_2O_3$ promotes the hydrolysis of slag, thereby promoting the hydration of slag. However, when excessive gypsum is added, too much AFt is generated in the hardened paste, and the expansion of AFt destroys the microstructure of geopolymer concrete, causing a decrease in compressive strength.

When NaCl and gypsum are combined, the surface chart shows a double-peak type. Both NaCl and gypsum can improve the 3-day compressive strength of geopolymer concrete, while the 3-day compressive strength is only slightly improved by NaCl at low dosage. NaCl can react with C-A-H and AFm to form Friedel's salt, thus reducing the formation of AFt. For the microscopic properties of Friedel's salt is not as good as AFt, the improving effect of gypsum on early strength is weakened, resulting in a concave chart in Figure 7 when NaCl content is 2%. When NaCl is increased to 4%, the surface chart shows a convex shape, which reflects the effect of NaCl on improving early strength. With the increase in gypsum, the 3-day compressive strength of geopolymer concrete first increases, then decreases, and then increases to 26.6 MPa when gypsum is 10%. When NaCl is increased to 6%, the 3-day compressive strength of geopolymer concrete showed the same changing trend as 4% NaCl and achieved a maximum value of 27.1 MPa at 5% gypsum. With the increase in NaCl, its contribution to the 3-day compressive strength is more obvious, and the optimal dosing of gypsum is also improved by NaCl. It is preliminarily predicted that the 28-day compressive strength is higher when the content of gypsum is 5–10% and the content of NaCl is 2–4%.

As shown in Figure 8, the curve rises slowly when NaCl is used alone. With the increase in NaCl, the 28-day compressive strength of geopolymer concrete only increases slightly. A 2% increase of NaCl improves the 28-day compressive strength from 14.2 MPa to 17.6 MPa, an increase of 23.9%. When gypsum is used alone, it shows a single-peaked curve; the 28-day compressive strength first increases and then decreases with the increase of gypsum. With 7.5% gypsum, the 28-day compressive strength improves from 14.2 MPa to 25.9 MPa, which is an increase of 82.3%. However, when too much gypsum is used, excess AFt is continuously generated in the hardened paste of geopolymer, and its expansion can destroy the hardened body structure and reduce the 28-day compressive strength [74]. For mortar, AFt is one of the main products that form the strength of the cementitious material. However, when too much gypsum is used, the compressive strength is reduced due to the expansion of AFt. The constraint effect of coarse aggregate is stronger than that of fine aggregate, i.e., more AFt can be accommodated in concrete, so the optimum dosage of gypsum in concrete is higher than that of mortar.

When NaCl and gypsum are combined, the surface chart shows a convex shape and obvious vertex appears. Combining 2% NaCl with 7.5% gypsum, the 28-day compressive strength reaches 28.6 MPa, which is 101.4%, 62.5%, and 10.4% higher than the OQ, L0, and 0C groups, respectively. When NaCl is increased to 4%, the 28-day compressive strength of the MC group reaches 34.1 MPa; when the dosage of gypsum is increased to 10%, the

28-day compressive strength of the MD group reaches the maximum value of 39.8 MPa; when NaCl is increased to 6%, the 28-day compressive strength decreases to 24.1 MPa in the HC group and 26.7 MPa in the HD group. The combination of NaCl and gypsum presents a superimposed effect that can solve the problems of insufficient strength and high shrinkage. It can be seen that the optimal dosage of gypsum increases as the dosage of NaCl increases. Gypsum provides $SO_4^{2-}$ for the gelling system, participates in early hydration, generates AFt, and improves early strength. NaCl reacts with C-A-H to form Friedel's salt; insoluble Kuzel's salt is formed due to the existence of gypsum, which contributes part of strength; part of $Cl^-$ is released in the formation of Kuzel's salt, and it continues to react with C-A-H to form Friedel's salt and simultaneously generates NaOH. Furthermore, NaOH promotes the hydration of slag and generates sodium-containing Zeolites. Due to the presence of calcium hydroxide, a small amount of calcium carbonate may also be formed [75,76]. $SO_4^{2-}$ is consumed with the hydration of slag going on, and AFt is transformed to AFm. Friedel's salt and Kuzel's salt may be generated under the influence of $Cl^-$. These cross-reaction products result in a synergistic effect on strength enhancement when NaCl and gypsum are used in combination [23,24,41], thus achieves higher strength. However, when there is too much NaCl, the strength of the geopolymer is reduced for the passivation effect of $Na^+$; excessive NaCl causes the solution to be saturated, and the remaining NaCl finally exists in the geopolymer system in the form of crystals. It not only hinders the migration of ions in the solution and restricts the occurrence of the polymerization reaction, it also weakens the cementing ability of geopolymer, increases the pore size distribution and pore volume of the sample, and then destroys the microstructure of the sample, which reduces strength.

The MC group with 4% NaCl and 7.5% gypsum, together with the MD group with 4% NaCl and 10% gypsum, are similar in compressive strength to the Reference group using 32.5 slag Portland cement, with both reaching the designed strength of 30 MPa. Gypsum can effectively assist quicklime in stimulating the activity of slag and improve the strength of geopolymer, and the addition of NaCl can make the strength of geopolymer concrete higher and more stable. Since many air-hardening cementitious materials such as quicklime and gypsum are used, it is necessary for the geopolymer concrete to be immersed in water to test the possible dissolution of dihydrate gypsum [77,78] and portlandite [79–81]. One of the main hydration products is AFt, but AFt is likely to be replaced and decomposed by sulfate and chloride [58,82–84]; therefore, sulfate immersion experiments are carried out to evaluate the effect on durability.

### 3.2. Effect of NaCl and Gypsum on Sulfate Resistance of Geopolymer Concrete

3.2.1. Strength under Water and Sulfate Immersion Environment

The compressive strengths of geopolymer concrete after immersing in water and sulfate solution for 0 days, 28 days, 56 days and 90 days are shown in Figures 9 and 10.

In Figure 9, as age increases, the strength of almost all test groups soaked in water increase, and the dark red color representing high strength gradually concentrates in the area with high NaCl and gypsum content. With the combined action of NaCl and gypsum, the strength increase improved by later hydration is much higher than the adverse effect of salt dissolution on strength. As the water immersion age increases, the strength of OQ group increases slightly compared with strength before water immersion. Because the pH value of the calcium hydroxide saturated solution is only 12.65 at 20 °C, the reaction activity of slag is not fully stimulated in a limited alkaline environment. After adding 7.5% gypsum, the compressive strength of the 0C group before immersion tests increases to 25.9 MPa, and the strength is also improved to 41.3 MPa, an increase of 59.5% after 56 days of soaking. However, after soaking for 90 days, the compressive strength is reduced to 36.8 MPa. Using gypsum alone improves the water immersion compressive strength at first but decreases strength as the immersion time increases. The remained gypsum generates AFt in the hardened paste with C-A-H, which destroys the structure of the geopolymer and reduces the strength. The water immersion strength of the L0 group increases slightly as the immersion time increases, but its compressive strength after 90 days of immersion is

not significantly higher than that of the OQ group. This indicates that when NaCl is used alone, although it can improve the strength at the initial stage of hydration, it has little effect on late strength after water immersion.

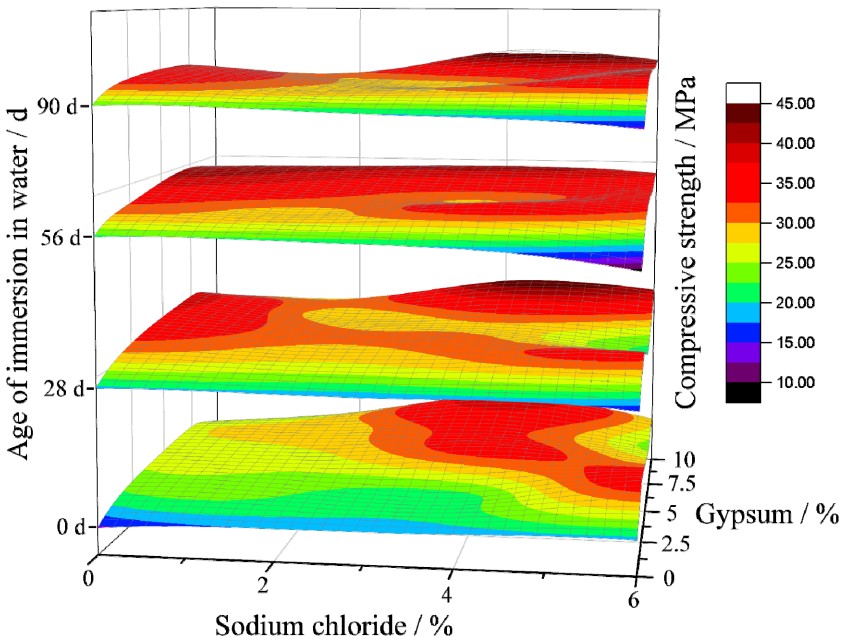

**Figure 9.** Water immersion strength.

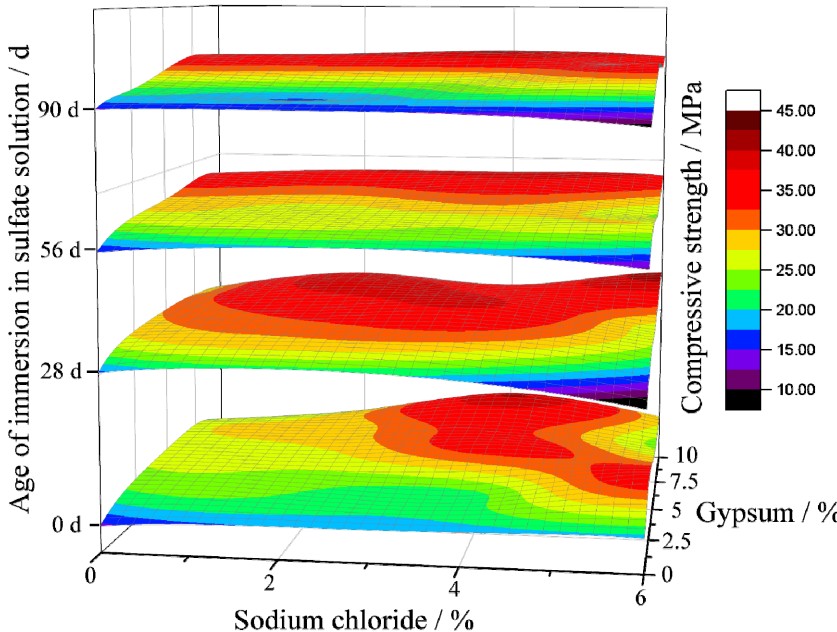

**Figure 10.** Sulfate immersion strength.

It is worth noting that when the dosage of NaCl is 6%, and the dosage of gypsum is 5~10%, the water immersion strength of the test groups HB, HC, and HD all show a large increase with age. Their compressive strength reaches about 39.0 MPa after immersing for 90 days; this indicates that high dosage of NaCl can continuously hydrate slag in water and increase the strength steadily with gypsum. The test group with 6% NaCl and 2.5% gypsum had little change in strength during water immersion, which indicates that sufficient gypsum is an essential element for the continuous hydration of geopolymers. Though the strength improvements of the groups with 6% NaCl and 5–10% gypsum are

higher, their 28-day compressive strengths fail to reach the design strength of 30 MPa. The compressive strengths of MC and MD groups not only reach the design strength at 28 days, they also increase with age while they are immersed in water. Finally, the compressive strengths reach 35.6 MPa and 46.1 MPa, respectively, after 90 days of immersion, and the strength of MD is close to the 48.7 MPa of the R group.

In Figure 10, the color of the area with higher NaCl content in the surface diagram turns dark red when soaked in sulfate solution for 28 days, and the strength of geopolymer concrete increases, indicating that NaCl can inhibit erosion caused by sulfate attack in the early stage. However, when immersed for 56 days and 90 days, the deep red color representing high strength is concentrated in the area of high gypsum content, indicating that the dosage of gypsum becomes the dominant factor affecting the sulfate erosion resistance of geopolymer. Meanwhile, 4% NaCl still has a certain effect on anti-sulfate erosion performance. In sodium sulfate solution, the strengths of most test groups increase after soaking for 28 days, and the maximum increase can reach 52.49%. Sodium sulfate is also a kind of activator for alkali-activated geopolymer, which can improve the hydration activity of slag and increase strength. However, it may cause excessive generation of AFt, and the expansion stress caused by the deposition of AFt may damage the concrete structure. After soaking in sulfate solution for 90 days, the strengths of groups LA, LB, MA, MB, and HA and groups L0 and OQ decrease, and the maximum strength loss reaches 25.21%. Low gypsum content does not improve the sulfate attack resistance of geopolymer concrete sufficiently. The strengths of 0C, LC, LD, MC, HB, HC, and HD groups all increase after soaking for 90 days, with the MC group reaching the highest strength of 39.5 MPa. In addition, the strength of the MC group increases steadily in sulfate solution, by 15.74%. High gypsum content can greatly improve the sulfate corrosion resistance of geopolymer concrete, but it may also have adverse effects when the content is too high.

The OQ group with neither NaCl nor gypsum was damaged seriously under sulfate attack. Its surface peeled off, and its strength was reduced. The erosion conditions of the L0 group and OQ group were similar, indicating that the single use of NaCl does not improve the sulfate erosion resistance of geopolymers significantly. The compressive strength of the 0C group increased gradually in sodium sulfate solution because the sodium sulfate in the solution enters the interior of the geopolymer concrete and promotes the hydration of the slag. This makes the concrete structure denser and inhibits the entry of subsequent erosion substances, thereby improving sulfate attack resistance of the geopolymer.

When 2% NaCl is added, the 90-day compressive strength of the test block in the sodium sulfate solution showed a trend of first increasing and then decreasing with increasing gypsum; it reached the maximum value of 36.6 MPa when the content of gypsum is 7.5%. When NaCl is increased to 4%, the 90-day compressive strength also shows a trend of first increasing and then decreasing with increasing gypsum, and it reaches the maximum value of 39.5 MPa in MC group. However, when NaCl is increased to 6%, the strength in sulfate solution is lower than that in water, and the 90-day compressive strength shows no obvious change with the increase of gypsum, indicating that too much NaCl can reduce the sulfate erosion resistance of the geopolymer concrete. To sum up, the appropriate dosage of NaCl is 4%. The strength of the MD group with 10% gypsum content decreases during sulfate immersion for 90 days, and the 90-day compressive strength is lower than that of the MC group, indicating that the appropriate dosage of gypsum is 7.5%. In conclusion, when the NaCl content is 4% and the gypsum content is 7.5% (group MC), the sulfate attack resistance of geopolymer concrete is at its best.

The group with the highest 90-day compressive strength in water (MD group) is different from that in sulfate solution (MC group). In fact, the MC group performs well under both conditions, reaching a strength of 35.6 MPa in water and 39.5 MPa in sulfate solution; this was the highest strength in the sulfate solution. These results show good resistance to sulfate attack. The corrosion resistance coefficient was calculated through the ratio of the compressive strength in sodium sulfate solution to the compressive strength

in water. The 90-day sulfate corrosion resistance coefficient of each test group is shown in Figure 11.

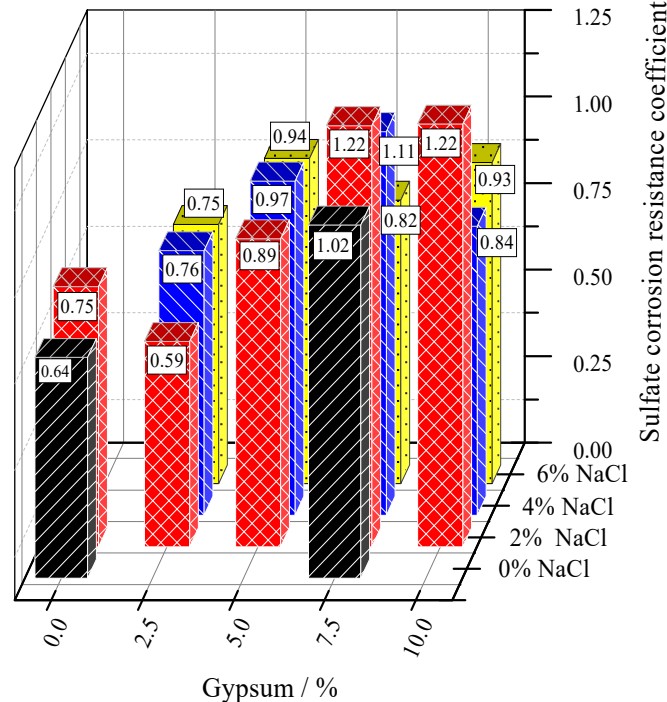

**Figure 11.** Sulfate corrosion resistance coefficient at 90 days.

The test shows that when immersed in 10% sodium sulfate solution for 90 days, the corrosion resistance coefficient of group R is only 0.8. The corrosion resistance coefficient of OQ and L0 groups are even lower than that of group R, which means their 90-day compressive strength when immersed in sulfate solution is significantly lower than that immersed in water, indicating poor resistance to sulfate erosion. When NaCl is used alone, the corrosion resistance coefficient of the L0 group increases by 17.2% compared with the OQ group; the gypsum alone increases the erosion resistance coefficient of the 0C group by 59.4% compared with OQ group. Both NaCl and gypsum can increase the corrosion resistance coefficient; in other words, they improve the sulfate erosion resistance of geopolymer concrete.

When NaCl and gypsum are combined, there is a synergistic effect on the corrosion resistance coefficient, and the corrosion resistance coefficient is further improved, but an overly high dosage of NaCl and gypsum also have adverse effects. When the NaCl content is 2%, as the gypsum content increases from 0 to 7.5%, the corrosion resistance coefficient also increases and reaches a maximum value of 1.22; however, it no longer increases when the gypsum content increases to 10%. When the content of NaCl is 4% and 6%, the corrosion resistance coefficient increases first and then decreases with the increase of gypsum. It reaches the maximum value of 1.11 and 0.94 with 7.5% and 5% gypsum, respectively. As the NaCl content increases, the maximum value of the corrosion resistance coefficient decreases. Although the corrosion resistance coefficient is higher when the NaCl dosage is 2%, the strength is higher when the NaCl dosage is 4%. When the gypsum dosage is 2.5%, 5%, and 7.5%, as the content of NaCl increases, the corrosion resistance coefficient first increases and then decreases. It reaches the maximum value when the NaCl dosage is 4%. However, the corrosion resistance coefficient of MD group is reduced for overly high content of gypsum. The corrosion resistance coefficient of MC group increases by 73.4% compared with that of the OQ group, and 38.8% higher than that of 32.5 slag cement. Both NaCl and gypsum can improve the sulfate attack resistance of quicklime-activated slag-based geopolymer concrete, and combined use of them can further improve the sulfate attack resistance.

The strength of MD group increases in water immersion test while decreases in sulfate immersion test, and its corrosion resistance coefficient is only 0.84. While the strength of MC group shows an increasing trend under the two conditions of water immersion and sulfate immersion, and its corrosion resistance coefficient reaches 1.11. In addition, the corrosion resistance coefficient of MC group is higher than that of MD and R groups, showing better sulfate erosion resistance performance. More importantly, the strength of MC is higher than that of 32.5 slag Portland cement after immersion in sulfate solution.

### 3.2.2. Mass Change

The mass changes of each group after immersing for 28 days, 56 days, and 90 days are shown in Figure 12.

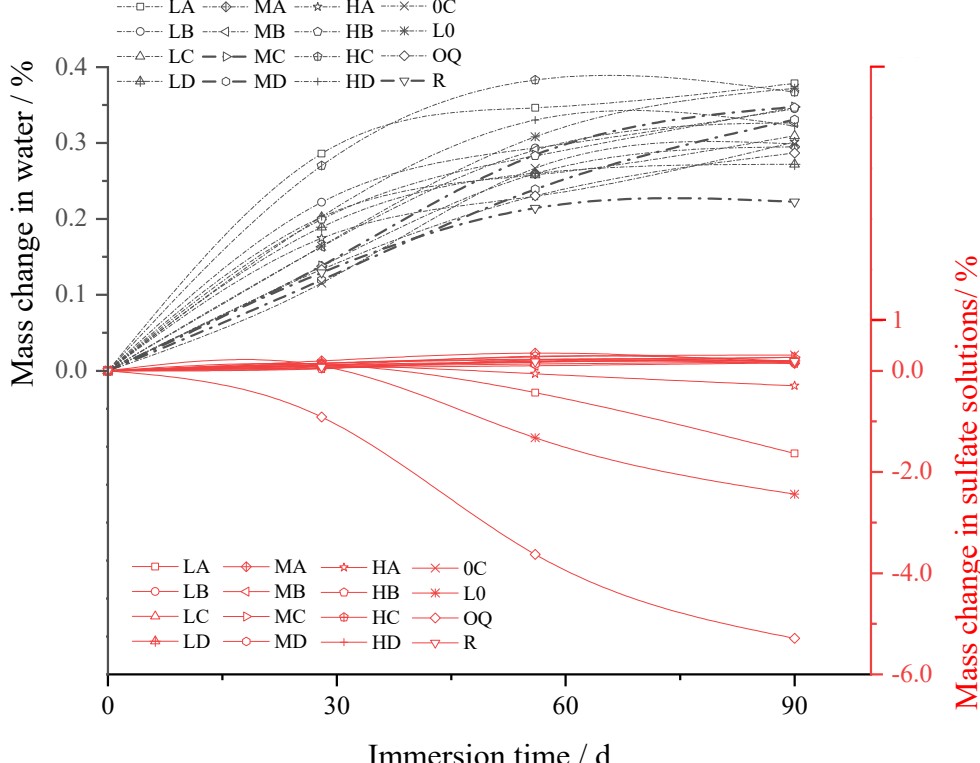

**Figure 12.** Mass change curve.

In Figure 12, all groups show increases in mass as they are immersed in water. Soaking in water is beneficial for the hydration of geopolymer. Sufficient water is conducive to the hydration reaction and generates more hydration products. Similar increasing trends are observed in mass and strength in each group, indicating that the added activators have fully reacted with slag, and no obvious erosion damage caused by the dissolution of soluble salt ions has occurred. After soaking in water for 90 days, the mass of group R, OQ, and L0 increase by 0.22%, 0.28%, and 0.37%, respectively. NaCl has a significant effect on mass increase but makes no major difference in strength after soaking in water. The mass of 0C group increased by 0.29%; although the strength of 0C group increased, it shows a trend of first increasing and then decreasing. Nevertheless, the mass of MC group continues to increase to 0.35%, and higher strength was obtained simultaneously. The mass of MD group increased by 0.33%, which was also the point when its strength was the highest. The combined effects of NaCl and gypsum promote hydration, leading to mass increase, therefore improving the strength of geopolymer concrete immersed in water.

The mass of the OQ and L0 group and the LA and HA group were significantly reduced when they were immersed in sulfate solution, and loose mud-like particles appeared on the surface. When immersing for 28 days, group OQ had up to 0.91% mass loss, and the mass

of the L0 group increased by 0.09%. After immersing for 90 days, OQ group had a mass loss of up to 5.29%, while the mass loss of the L0 group was only 2.44%. NaCl can improve the sulfate attack resistance of geopolymer concrete to a certain extent, but it cannot completely avoid the mass loss caused by sulfate attack. The mass of 0C group has always shown an increasing trend in 90 days, and its mass increases by 0.31%, indicating that gypsum can greatly improve the sulfate erosion resistance of geopolymer concrete. Under the combined action of NaCl and gypsum, the masses of LA, MA, and HA groups all show a trend of first increasing and then decreasing during the 90 days; the mass loss ratios of LA and HA groups is up to 1.63% and 0.30% at 90 days, respectively, while the mass of MA increases by 0.15%, which indicates that increasing the content of NaCl can also enhance the sulfate erosion resistance of geopolymer concrete. The sulfate corrosion resistance of MA group is improved compared with OQ and L0 groups, but there is still a tendency to lose mass, which indicates that when the gypsum content is low, the sulfate corrosion resistance of geopolymer concrete cannot fundamentally be improved. However, when gypsum is more than 5%, no matter how much NaCl is added, no mass loss occurred at all. The effect of gypsum on improving the sulfate erosion resistance of the geopolymer concrete is quite significant. Under the combined action of NaCl and more than 5% gypsum, the masses of LC, MC, and MD groups increase by 0.26%, 0.27%, and 0.17%, respectively, and the mass of the MC group increases the most; its strength increases steadily in the sulfate solution. The combined effects of NaCl and gypsum can not only avoid the mass loss caused by sulfate attack, but also obtains higher and more stable strength.

### 3.2.3. Change of Relative Dynamic Elastic Modulus

The relative dynamic elastic modulus change of geopolymer concrete under water immersion and sulfate solution erosion is shown in Figure 13.

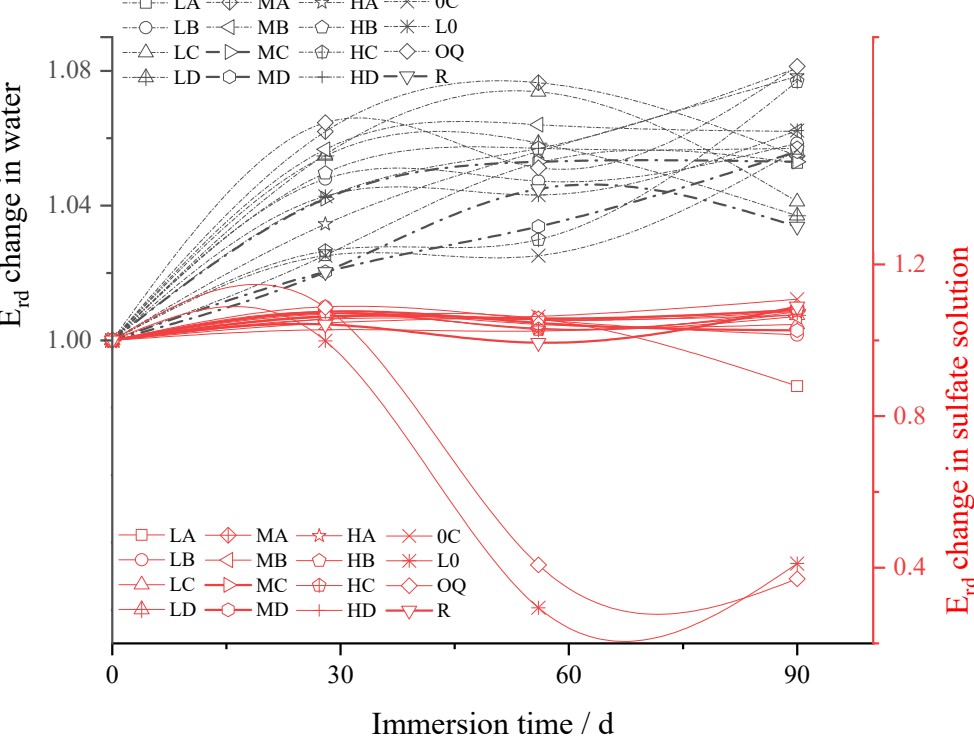

**Figure 13.** Change of relative dynamic elastic modulus.

As can be seen from Figure 13, the relative dynamic elastic modulus of each test group immersed in water has increased in 90 days but exhibits different trends. The relative dynamic elastic modulus of group OQ increases at 28 days, decreases at 56 days, and increases again at 90 days; the relative dynamic elastic modulus of group 0C and

L0 increases at 28 days, changes little at 56 days, and continues to increase at 90 days; both sodium chloride and gypsum contribute to the increasing of relative dynamic elastic modulus. The relative dynamic elastic modulus of LA and LB groups increase, while the relative dynamic elastic modulus of LC and LD groups increase first and then decrease, indicating that high gypsum content is not conducive to the stable growth of the relative dynamic elastic modulus under low sodium chloride content. When sodium chloride increases to 4%, the relative dynamic elastic modulus of MA group increases first and then decreases; the relative dynamic elastic modulus of MB group increases at 28 days and remains stable then; the relative dynamic modulus of MC and MD groups increase significantly at 28 days, reaching 1.04 and 1.02, then increase to 1.05 and 1.06 at 90 days; this indicates that the appropriate dosage of gypsum increases with the increase of sodium chloride, and the relative dynamic elastic modulus can no longer increase stably with low dosage of gypsum, but can increase stably with high dosage of gypsum. When the dosage of NaCl increases to 6%, the relative dynamic elastic modulus of HA, HB, HC, and HD groups are not obviously changed as gypsum increases. NaCl plays a leading role at high dosage, and the relative dynamic elastic modulus all increase to nearly 1.08. The combined use of NaCl and gypsum improves the relative dynamic elastic modulus of geopolymer concrete soaked in water steadily, and the effect of NaCl is particularly significant.

As shown in Figure 13, the relative dynamic elastic modulus curves of some test groups, such as OQ, L0, and LA, in sodium sulfate solution show a significant decrease. It can be seen that the relative dynamic elastic modulus of the OQ group increases to 1.09 at 28 days, then decreases significantly to 0.37 at 90 days; the L0 group also shows a similar trend, and the relative dynamic elastic modulus decreases to 0.41 at 90 days. NaCl only slightly prevents the decrease of relative dynamic elastic modulus. The relative dynamic elastic modulus of group 0C continues to increase to 1.11 at 90 days, indicating that gypsum plays a key role in improving the sulfate attack resistance of geopolymer concrete. However, due to the expansion effect of Aft, the compressive strength of group 0C in the water immersion test first increases and then decreases, showing unstable performance in the latter period. After 90 days of immersion in sodium sulfate solution, the relative dynamic elastic modulus of the LC group is up to 1.04; the relative dynamic elastic modulus of LD group remains the same as LC. Excessive gypsum content does not increase the relative dynamic elastic modulus. However, the relative dynamic elastic modulus of LA group decreases to 0.88 at 90 days. The relative dynamic elastic modulus cannot be fully improved when the gypsum content is insufficient. When NaCl increases to 4%, the relative dynamic elastic modulus of the MC group is further increased to 1.08 after soaking in sulfate salt for 90 days, showing that NaCl helps to improve the relative dynamic elastic modulus. When gypsum increases to 10%, the relative dynamic elastic modulus decreases to 1.03, indicating that too much gypsum can decrease the relative dynamic elastic modulus. Neither too low NaCl content nor too high gypsum content is conducive to the stable development of the relative dynamic elastic modulus. The relative dynamic elastic modulus of MC group rises steadily to 1.09 within 90 days, which is the same as that of group R. The MC group produces more hydration products through the combined action of NaCl and gypsum. The microstructure of geopolymer concrete is also denser, which makes the relative dynamic elastic modulus increase steadily.

### 3.2.4. Sulfate Attack Products

Corrosion products analysis were carried out with X-ray diffraction (XRD) on the OQ and L0 groups with serious erosion damage after 90 days of sulfate immersion, the 0C, LC, and MC groups with better sulfate resistance, and the MD group with reduced strength in sulfate solution. The results are shown in Figure 14; on both sides of Figure 14 are the surface condition of the test cubes after 90 days of sulfate immersion. Meanwhile, the morphologies of erosion products were observed by scanning electron microscopy (SEM), as shown in Figures 15–17.

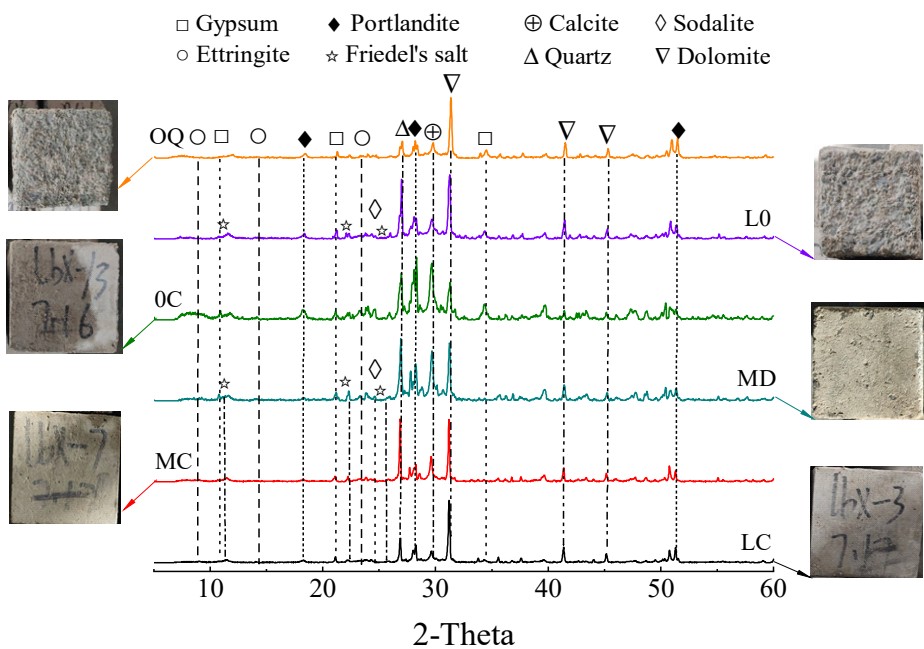

**Figure 14.** XRD analysis of sulfate attack products.

L0(×10,000)

OQ(×10,000)

(**a**) Water immersion

L0(×10,000)

OQ(×10,000)

(**b**) Sulfate immersion

**Figure 15.** SEM photos of Group L0 and OQ's erosion products.

MD(×10,000)

0C(×10,000)

(**a**) Water immersion

MD(×10,000)

0C(×10,000)

(**b**) Sulfate immersion

**Figure 16.** SEM photos of Group MD and 0C's erosion products.

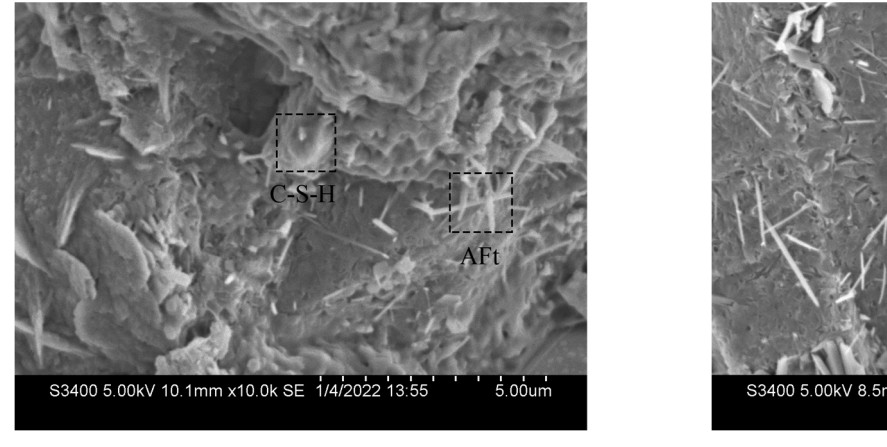

(**a**) Water immersion (×10,000)

(**b**) Sulfate immersion (×10,000)

**Figure 17.** SEM photos of MC.

In Figure 14, after soaking in sulfate solution for 90 days, the diffraction peaks of gypsum and ettringite appear in the XRD patterns of groups OQ and L0, and their surfaces are granulated and peeled off severely. It can be seen in Figure 15a that the products of the OQ and L0 groups immersed in water are mainly C-S-H and C-A-H gels and a small

amount of hexagonally shaped calcium hydroxide [85]. Calcium hydroxide is reduced due to the reaction with slag, making it difficult to observe. While in Figure 15b, obvious needle-like AFt crystals [86] and needle-like, flaky, dihydrate gypsum crystals can be observed. The OQ and L0 groups were not damaged when soaked in water, but serious ettringite crystal erosion and gypsum crystal erosion occurred in OQ and L0 groups soaked in the sulfate solution. Remaining $Ca(OH)_2$ reacts with $SO_4^{2-}$ in sodium sulfate solution to form gypsum, and ettringite is produced under the reaction of C-A-H with $SO_4^{2-}$ in the solution. In addition, the diffraction peaks of Friedel's salt and sodalite also exist in L0, LD, MC, and MD groups, as shown in Figure 14. NaCl reacts with C-A-H to form Friedel's salt and generates NaOH; NaCl also reacts with NaOH and active $SiO_2$, $Al_2O_3$, producing sodalite ($Na_4Al_3Si_3O_{12}Cl$). A small amount of Kuzel's salt is also produced in low concentration of free chloride ions. As a result of its low content, no obvious diffraction peaks of Kuzel's salt are observed in the XRD pattern. The hardened paste samples still contain a small number of fine aggregate components, so the characteristic peaks of quartz, dolomite, and calcite appear in the XRD pattern. The diffraction peaks of gypsum and ettringite appear in 0C, MD, and MC. As shown in Figures 16 and 17, needle-like and plate-like crystals are observed in both water-soaked and sulfate-soaked test groups of 0C, MC, and MD. Because gypsum and ettringite are hydration products, it is not possible to use only XRD patterns to accurately judge whether gypsum and ettringite are generated for sulfate attack.

As for the strength change of group 0C that increases first and then decreases during immersion in water, a large number of needle-like AFt crystals observed in Figure 16a 0C may explain this. The continuous increase of AFt makes the structure denser at first, but its expansion effect can destroy the microstructure of geopolymer concrete when it is excessive. The strength of group 0C is improved in sulfate solution, for sodium sulfate promotes the continuous hydration of slag, which plays a key role in the repair of geopolymer concrete structure. The strength of the MC group increases steadily in the sodium sulfate solution, and the 90-day compressive strength is higher than when it is soaked in water, indicating that no obvious erosion damage occurred in the MC group. However, the strength of the MD group first decreases and then increases in the sulfate solution. The 90-day compressive strength of the MD group is also lower than that soaked in water. It can be confirmed that erosion damage occurred in the MD group during the sulfate immersion test.

The product morphologies of the MC group In water and sulfate solution are shown in Figure 17. It can be seen that there are fewer calcium hydroxide flaky crystals in MC group soaked in water, because calcium hydroxide is more involved in hydration reaction. The MC group is less affected by sulfate attack in the sulfate solution due to the compact structure.

The unreacted $Ca(OH)_2$ in the geopolymer undergoes the following reaction in the sodium sulfate solution: $Ca(OH)_2 + Na_2SO_4 + 2H_2O = CaSO_4 \cdot 2H_2O + 2NaOH$ [87]. Since the solubility of $CaSO_4$ is less than that of $Ca(OH)_2$, the reaction proceeds in the positive direction. The generated NaOH promotes the dissolution of unreacted slag to form C-S-H and C-A-H. Then C-A-H reacts with $SO_4^{2-}$ to form aFt. Hence, there must be ettringite and gypsum produced by the sulfate attack in 0C, MC, and MD. The degree of slag hydration varies for different dosages of NaCl and gypsum. Therefore, the amount of unreacted $Ca(OH)_2$ differs, thus leading to different properties of geopolymer concrete, causing the strength increase of MC and the strength decrease of MD.

Calcium-rich alkali-activated geopolymers, due to the formation of AFt and gypsum, generally have low sulfate erosion resistance [88–91], while the calcium-rich system geopolymer made of NaCl, gypsum, quicklime and slag has a good anti-sulfate attack effect. This effect is due to the conduction of calcium to Friedel's salt and Kuzel's salt. In the MC and MD groups, NaCl and gypsum promote the hydration of slag, consume calcium hydroxide, and inhibit the occurrence of sulfate erosion. $Cl^-$ can react with C-A-H and AFm in the hardened paste to form Friedel's salt ($3CaO \cdot Al_2O_3 \cdot CaCl_2 \cdot 10H_2O$) [47,92,93]. This form of combined chloride ions can not only refine the pore size, but also effectively increase the tortuosity of the pore network system [42], improve the sulfate erosion resistance of geopolymer concrete, and reduce the intrusion of $SO_4^{2-}$. Due to the low concentration of

$Cl^-$, Kuzel's salt ($3CaO \cdot Al_2O_3 \cdot 0.5CaSO_4 \cdot 0.5Ca\ Cl_2 \cdot 10H_2O$) is generated with $SO_4^{2-}$, filling pores and contributing to strength [94,95].

## 4. Conclusions

Based on the above experimental studies, the following conclusions can be drawn:

(1) The slag cannot be fully hydrated when activated by quicklime only; under that condition, its 3-day compressive strength is only 8.6 MPa, and the 28-day compressive strength is only 14.2 MPa. When 2% NaCl is used alone, the polycondensation of silicon–aluminum–oxygen tetrahedral units is promoted by NaCl, sodalite is produced, and the 3-day compressive strength improves to 13.4 MPa, representing a 55.8% increase. Its 28-day compressive strength improves to 17.6 MPa, representing a 23.9% increase. When 7.5% gypsum is used alone, a large amount of AFt is generated in the early stage of hydration. In addition, the consumption of active $Al_2O_3$ promotes the hydrolysis of slag, thereby promoting the hydration of slag. The 3-day compressive strength improves to 29.7 MPa, representing a 245.3% increase. The 28-day compressive strength is reduced to 25.9 MPa because too much AFt is generated in the hardened paste.

(2) The combination of NaCl and gypsum presents a superimposed effect that can solve the problem of insufficient strength. When 4% NaCl and 10% gypsum are combined, the 3-day and 28-day compressive strength are increased to 26.6 MPa and 39.8 MPa, respectively, reaching a similar strength as R group. The optimal dosage of gypsum improves as the dosage of NaCl increases. Gypsum provides $SO_4^{2-}$ and participates in early hydration, generates AFt, and improves early strength. With NaCl and gypsum, Friedel's salt and insoluble Kuzel's salt are generated, and they generate NaOH at the same time. NaOH can further promote the hydration of slag and generate sodium-containing zeolites. These cross-reaction products result in a synergistic effect on strength enhancement.

(3) Water immersion shows a positive effect on the composite-activated geopolymer. Most of the soluble salt ions are involved in the reaction, and no obvious corrosion caused by the dissolution of salt activator occurred. Both NaCl and gypsum can improve the sulfate resistance of geopolymer concrete, and when combined, the sulfate corrosion resistance coefficient is improved by 73.4%, which is even higher than that of 32.5 slag Portland cement. NaCl and gypsum can promote the hydration of slag and accelerate the consumption of calcium hydroxide, thus reducing the sulfate erosion caused by calcium hydroxide. In the early stage of hydration, the expansion of Aft and the filling effect of Friedel's salt can also make the geopolymer denser and enhance the sulfate resistance of geopolymer. The formation of Kuzel's salt consumes the invaded $SO_4^{2-}$ and inhibits the formation of ettringite in the hardened paste, thereby suppressing sulfate attack damage. The conduction of calcium to Friedel's salt and Kuzel's salt improves the sulfate resistance greatly.

(4) The slag-based geopolymer activated by NaCl, gypsum, and quicklime has the advantages of low energy consumption, low carbon emission, and low cost. Research on reinforcement corrosion in the geopolymer concrete has not yet been completed, so its application is presently limited to plain concrete engineering, such as curb stone. It can also be used as a curing agent to solidify coastal saline soil, and related research has been conducted.

**Author Contributions:** Conceptualization, W.H. and X.M.; methodology, W.H. and Q.S.; validation, W.H. and Q.S.; formal analysis, B.L. and Q.S.; investigation, B.L. and Q.S.; data curation, B.L. and Q.S.; writing—original draft preparation, B.L. and X.M.; funding acquisition, W.H. and Q.S. All authors have read and agreed to the published version of the manuscript.

**Funding:** This work is sponsored by National Natural Science Foundation of China (No. 52208413), National Key R&D Program of China (2022YFC3803400), Beijing Municipal Natural Science Foundation (No. 8204058), BUCEA Post Graduate Innovation Project (No. 02081022003) and the R&D Program of Beijing Municipal Education Commission (No. KM202210016011).

**Institutional Review Board Statement:** Not applicable.

**Informed Consent Statement:** Not applicable.

**Data Availability Statement:** The data used to support the findings of this study are available from the corresponding author upon request.

**Conflicts of Interest:** The authors declare no conflict of interest.

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
