# Peer review of "Compound Effects of Sodium Chloride and Gypsum on the Compressive Strength and Sulfate Resistance of Slag-Based Geopolymer Concrete"

_buildings, doi:10.3390/buildings13030675_

Round 1

Reviewer 1 Report

- the observed 28 d activity coefficient was 95, 95%?

- Table 1, Cao should be “CaO”.

-building gypsum is CaSO4.0.5H2O?

- the activated effect of NaCl to GBFS (slag in the study) in known. However, the higher Cl content in inner cementitious materials is a great risk for the concrete containing steel bar or the steel structure. So, I suggest that the authors should provide an assessment for the application of the slag-based geopolymer concrete containing NaCl and gypsum in part of Conclusion.

Reviewer 2 Report

This paper presents some interesting information on the compound influence of sodium chloride and gypsum on the sulfate resistance of slag-based geopolymer concrete.

Technically, the research plan is good and initially clarify the role of sodium chloride and gypsum in geopolymer concrete. The paper could be published after minor revisions. Some suggested amendments are listed below.

1. Pg.1 Line 18: Change "can further increased" to "can be further increased"

2. Pg.1 Line 30, 36: Change "," to ",".

3. Pg.3 Line 91: "Mixed water for municipal tap water" should be deleted, as the mixed water is detailly described later.

4. Pg.4 Tab.4: "Reference group" was used in Table 4, but in the other place "R group" was used. I suggest using the same word.

5. Pg.5 Line 156~157: Change "SO42-" to "SO42-".

6. Pg.6 180~196: The unit format in the equations should be consistent.

7. Pg.8 Line 253: Change "with the increase of" to "With the increase of".

8. Pg.9 Fig.4, Fig.5: The x, y-axis tick marks can’t be clearly identified, it should be improved.

9. Pg.13 Fig.8: The max value of the ordinate axis should be marked, and its name should be " Erd ", not "Erd", as the author defined in Line 181.

10. Pg.14 508, Pg.16 548: Change "AFT" to "AFt".

11. Pg.15~16 Fig.10~12: The layout of those figures should be further optimized. Some of the figures and their names are not in the same page. The magnification of the SEM images should be marked.

12. Both compressive strength and sulfate resistance are studied in this paper, maybe the article title should be replaced by "Compound effects of sodium chloride and gypsum on the compressive strength and sulfate resistance of slag-based geopolymer concrete".

Reviewer 3 Report

Dear Authors,

Overall, the manuscript is interesting in terms of topic. However, The manuscript needs to be checked for spelling and grammar. there are a lot of typos and grammatical issues in the text. some of the statements needs reference(s) and supports.  Please find my detailed technical comments attached. 

Thanks!

Reviewer 4 Report

Interesting and important research. Nice manuscript. Congratulations to the authors. In my opinion, for a scientific article, there are too many citations of norms / technical standards.

Reviewer 5 Report

The article studied the effect and mechanism of NaCl and gypsum on the compressive strength, the resistance to water, and the sulfate attack of geopolymers. The experimental activity was well-conceived and performed correctly. Nevertheless, some issues are not correctly addressed, and the article needs revisions before being suitable for publication. Some detailed comments are provided below:

·       Abstract: The authors should state the materials and methods used in the study. This abstract only contains aim, results and conclusions.

·       Lines 65-73: The gap in the paper is not clear. The authors should state what is done and what is not in the literature as well as the novelty of the publication.

·       Lines 89-92: What is the absorption of the aggregates? This is a key property, so the authors should state them.

·       Line 136: There is something wrong with the references.

·       Lines 149-160: What is the mix proportioning of the aggregates? The authors did not state this in the article.

·       Lines 161: I believe that the compressive strength after immersion should be after 90d, right? It does not really make sense to do it at 0d since it is the same as before (28d).

·        Lines 210: How did the authors obtain the intermediate values? Are they obtained by ANOVA or any similar tool? Please explain this.

·       Lines 435-436: I am not sure about that sentence. Soaking geopolymers for longer times (after 28d) is not beneficial for hydration. Did the authors find this in any reference? Please explain why.

·       Lines 605-620: I feel that this is completely unconnected with the main focus of the article. I would remove this section or connect it in a better way.

·       Lines 621-665: The conclusions are too long. They are an overview of the results instead of the findings of the paper. Please revise them.

Reviewer 6 Report

This paper investigates the compound effects of sodium chloride and gypsum on the sulfate resistance on geopolymer concrete. The research topic is exciting. However, it must address the following comments to improve the quality of the manuscript further:

(1) Abstract. It is recommended to add the quantified research findings in a more consise form.

(2) Introduction and literature review. There are many types of materials ans nanomaterials, often used in combination, in orded to improve secific parameters of concrete. Why did the authors select sodium chloride and gypsum? Some illustration should be added, and some new references from this topic can be used:

- Mechanical Performance of Date-Palm-Fiber-Reinforced Concrete Containing Silica Fume, Buildings 2022.

- „Research on the fracture mechanical performance of basalt fiber nano-CaCO3 concrete based on DIC technology, Construction and Building Materials 2022.

- “Combined effect of coal fly ash (CFA) and nanosilica (nS) on the strength parameters and microstructural properties of eco-friendly concrete”, Energies 2023.

(3) Materials. Please provide apprereance or SEM photos or materials used.

(4) Activities. Please provide flow-chart from the studies.

(5) Methods. Please provide photos showing activities desribed in section 2.2., i.e. mix design, sulphate attack, etc.  This part is very interesting and should be better presented.

(6) SEM photos. Please describe phases visible on SEM photos. How the structure of the matrix changed after the sulfate attack and the remedial actions taken.

(7) Conclusions. Please emphaize the main innovaltive findings from the studies more sharply. In its current form, the conclusions are too long-winded.

Round 2

Reviewer 3 Report

Dear Authors,

The manuscript looks fine and the comments were addressed. 

Thanks,

Reviewer 5 Report

The authors did not address most of my comments, such as:

- Including the materials and methods in the abstract.

- Specifying the aggregate proportions

- Including the method they used to obtain the intermediate values in the manuscript.

- Etc.

Because of this, I would reject this article.

Reviewer 6 Report

The paper is well revised. I have no further comments.